

# A digital mapping application for quantifying and displaying air temperatures at high spatiotemporal resolutions in near real-time across Australia

Mathew Webb and Budiman Minasny

School of Life and Environmental Sciences & Sydney Institute of Agriculture, University of Sydney, Eveleigh, NSW, Australia

## ABSTRACT

Surface air temperature ($T_a$) required for real-time environmental modelling applications should be spatially quantified to capture the nuances of local-scale climates. This study created near real-time air temperature maps at a high spatial resolution across Australia. This mapping is achieved using the thin plate spline interpolation in concert with a digital elevation model and 'live' recordings garnered from 534 telemetered Australian Bureau of Meteorology automatic weather station (AWS) sites. The interpolation was assessed using cross-validation analysis in a 1-year period using 30-min interval observation. This was then applied to a fully automated mapping system—based in the R programming language—to produce near real-time maps at sub-hourly intervals. The cross-validation analysis revealed broad similarities across the seasons with mean-absolute error ranging from 1.2 °C (autumn and summer) to 1.3 °C (winter and spring), and corresponding root-mean-square error in the range 1.6 °C to 1.7 °C. The $R^2$ and concordance correlation coefficient ($P_c$) values were also above 0.8 in each season indicating predictions were strongly correlated to the validation data. On an hourly basis, errors tended to be highest during the late afternoons in spring and summer from 3 pm to 6 pm, particularly for the coastal areas of Western Australia. The mapping system was trialled over a 21-day period from 1 June 2020 to 21 June 2020 with majority of maps completed within 28-min of AWS site observations being recorded. All outputs were displayed in a web mapping application to exemplify a real-time application of the outputs. This study found that the methods employed would be highly suited for similar applications requiring real-time processing and delivery of climate data at high spatiotemporal resolutions across a considerably large land mass.

# INTRODUCTION

A timely and accurate source of air temperature ($T_a$) data is essential for a wide variety of environmental modelling applications requiring real-time monitoring of environmental

Corresponding author
Mathew Webb,
mathew.webb@dpipwe.tas.gov.au

change (*Lazzarini et al., 2014*). This is often gleaned from a network of in situ telemetered meteorological weather stations that are streamed over the internet (*Williams et al., 2011*). However, datasets of this nature tend to be relevant for a single geographic location that fail to accurately account for the spatial variability between sites that can vary markedly over short distances (*Webb et al., 2016*). For applications that rely on location-specific data, observations are often harvested from stations situated kilometres away from their location of interest, resulting in that data not being truly representative of the desired location (*Jeffrey et al., 2001*; *Liu et al., 2018b*). Thus, $T_a$ can vary considerably over space and time, often attributed to the effects of topographic, coastal and latitudinal factors (*Hutchinson, 1991*; *Jarvis & Stuart, 2001a*; *Wang et al., 2011*), cloud cover (*Xue et al., 2019*), radiative effects from aerosols (*Li et al., 2017*; *Mitchell et al., 1995*) and diurnal variation (*Liu et al., 2018a*). As such, $T_a$ for the purpose of input to real-time modelling applications need to be spatially quantified to dynamically account for these interactions but also at an appropriate spatial and temporal resolution to account for the subtle nuances of local-scale climates.

There has been a plethora of research aimed at interpolating surface air temperature at various spatiotemporal scales (*Hutchinson, 1991*; *Jarvis & Stuart, 2001b*; *Jeffrey et al., 2001*; *Jones, Wang & Fawcett, 2009*; *Xu et al., 2018*). This is in addition to surface temperature estimated from satellite data (*Mao et al., 2017*; *Sobrino, Julien & García-Monteiro, 2020*). Or from regional reanalysis of global circulation models at high spatiotemporal resolutions (*Bollmeyer et al., 2015*; *Su et al., 2019*). Despite this, their application to real-time monitoring of climate have been limited, or insufficient for local-scale monitoring purposes. For example a modelling system based on remote sensing data coupled with in situ meteorological recordings was able to produce air temperature maps in near real-time across the United Arab Emirates (*Lazzarini et al., 2014*). However, the spatial resolution of ~3 km was limited in accounting for lapse rates in highly variable topography, despite the system capable of delivering outputs at very high temporal resolution (every 15-min). Similarly, a near real-time drought monitoring tool developed for South Asia (*Aadhar & Mishra, 2017*), capable of producing daily minimum and maximum temperatures at a spatial resolution of 0.05° (~5 km), would also require further adaptation for high resolution monitoring. This is in addition to a similar system currently used in Australia, where daily minimum ($T_{min}$) and maximum ($T_{max}$) temperatures are produced from Australian Bureau of Meteorology (BoM) weather station sites using thin plate smoothing splines (TPS) interpolation to deliver daily products at 0.05° (~5 km) grid resolution (*Jeffrey et al., 2001*; *Jones, Wang & Fawcett, 2009*). While both datasets are useful for broad-scale analysis requiring up-to-date daily records, they still lacked the resolution for sub-daily real-time monitoring at the local-scale.

Recently, a near-real time mapping system was developed to produce air temperature maps at high spatiotemporal resolutions across the state of Tasmania, Australia (*Webb, Kidd & Minasny, 2020*). This used a combination of regression trees (RT) and TPS interpolation and capable of consistently producing maps at a spatial resolution of 80 m at 1-h time steps. Evaluation of the system showed that the TPS method was highly suited to real-time application due to the speed and relative accuracy of the outputs produced.
For example assessment of the TPS interpolation showed root mean square errors were consistently under 1.5 °C, in addition to only requiring 2-min processing time to produce each map product. In this context, the application would be suited to the estimation of $T_a$ across a much larger geographic space at a similar spatiotemporal resolution. As such, there is also an opportunity to apply this approach on a digital platform for real-time access for end-users.

The objective of this study was to apply and extend the methods in *Webb, Kidd & Minasny (2020)* for production of $T_a$ maps across continental Australia. TPS interpolation is used to produce $T_a$ maps at sub-hourly intervals (every 30-min) based on recordings garnered directly from BoM automatic weather station (AWS) sites. The resulting maps are presented digitally at a spatial resolution of 286 m, appropriate for local-scale monitoring purposes. The methods for prediction accuracy are evaluated using historic hourly $T_a$ data captured over a 1-year period, in addition to assessing the efficacy of the system for real-time application and subsequent display of outputs in a purpose-built web mapping application.

## MATERIALS AND METHODS

### Approach

The present study consisted of 2 parts. Firstly, evaluation of the TPS methodology using cross-validation; and secondly, application of the methodology for operational real-time mapping of $T_a$ (Fig. 1). For the evaluation purpose of the study, a historical dataset of 30-min interval $T_a$ recordings was garnered from BoM AWS sites for the 1-year period 1 March 2019 to 29 February 2020. This data was used in a leave-one-out cross-validation exercise to assess the prediction performance of the TPS interpolation method. For the application of the methodology for operational real-time mapping, this was tested over a 21-day period from 1 June 2020 to 21 June 2020. For this purpose, a fully automated mapping system was developed using R programming language (*R Development Core Team, 2015*). Processing performance of this mapping system was evaluated for computational efficiency by analysing each subsequent spatial output (i.e. the time to taken to produce each $T_a$ map) and therefore assessed for real-time application. Maps produced from the interpolation process are immediately displayed in a web map application.

### Air temperature data

Air temperature data recorded by AWS from the BoM and capable of providing real-time access at 30-min intervals were considered for primary use in this study (Fig. 2). For evaluating the accuracy of the model, a requirement was set, where each station used for the real-time application should have historic recordings for the previous year, specifically from 1 March 2019 to 29 February 2020. These historical data were used for cross-validation analysis. It should be noted that not all AWS sites had data available for the entire evaluation period. Thus, AWS sites that had least 15 days of recordings—consisting of 30-min interval recordings in each season—was considered for the evaluation process. AWS sites that did not meet this criterion were discarded from the analysis (192 in total). Thus, the screening process resulted in 534 AWS sites corresponding to a

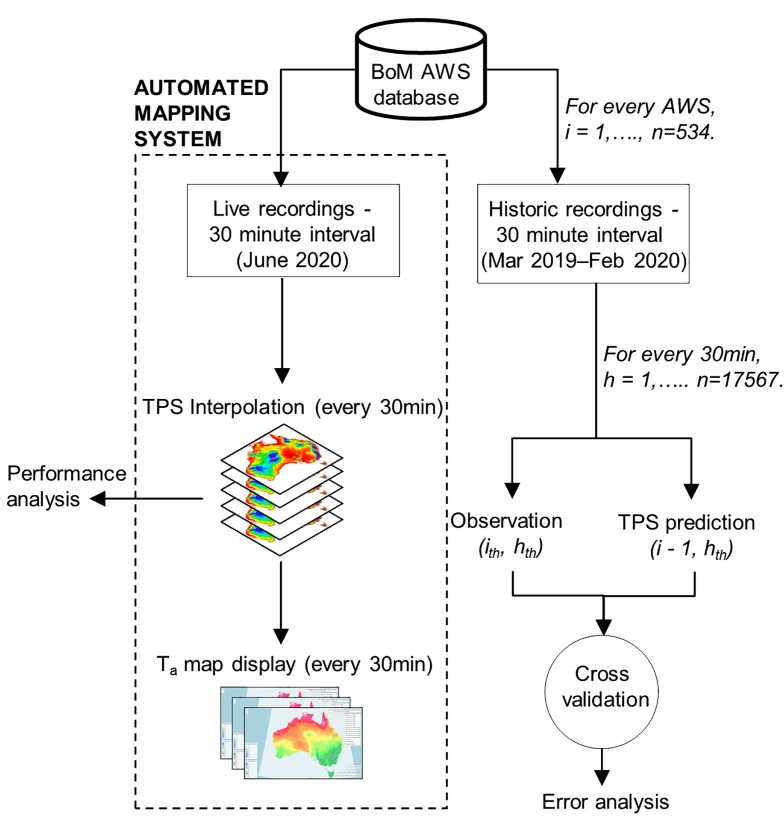

**Figure 1 Workflow developed for this study.**

possible 17,567 recording observations in the evaluation period and relevant to each AWS. It should be noted that AWS air temperature observations are recorded using a resistance temperature detector placed within a Stevenson weather screen at 1.2 m above ground (*Bureau of Meteorology, 2018*). All AWS recordings are telemetered into the BoM climate database and publicly accessible via URL (http://www.bom.gov.au/tas/observations/). These are typically displayed at 30-min intervals. However, due to telemetry and processing delays, readings tend to lag the true observation time of approximately 10- to 20-min.

## Interpolating $T_a$ using thin plate smoothing splines

Air temperature values garnered from the BoM AWS sites were interpolated on a 30-min interval basis using TPS. This was performed to form TPS predictions in the evaluation period (1 March 2019 to 29 February 2020) as well as for application to real-time mapping. Its application involves a trivariate approach whereby latitude, longitude, and elevation variables are used as independent variables, as per *Jeffrey et al. (2001)*. The independent variables of latitude and longitude are used for the partial spline component to account for spatial variation, whereas elevation is combined to account for the temperature lapse rates. The spline component of the algorithm is optimised by minimising the generalised cross validation error from the residual sum of squares (*Hutchinson, 1991*).

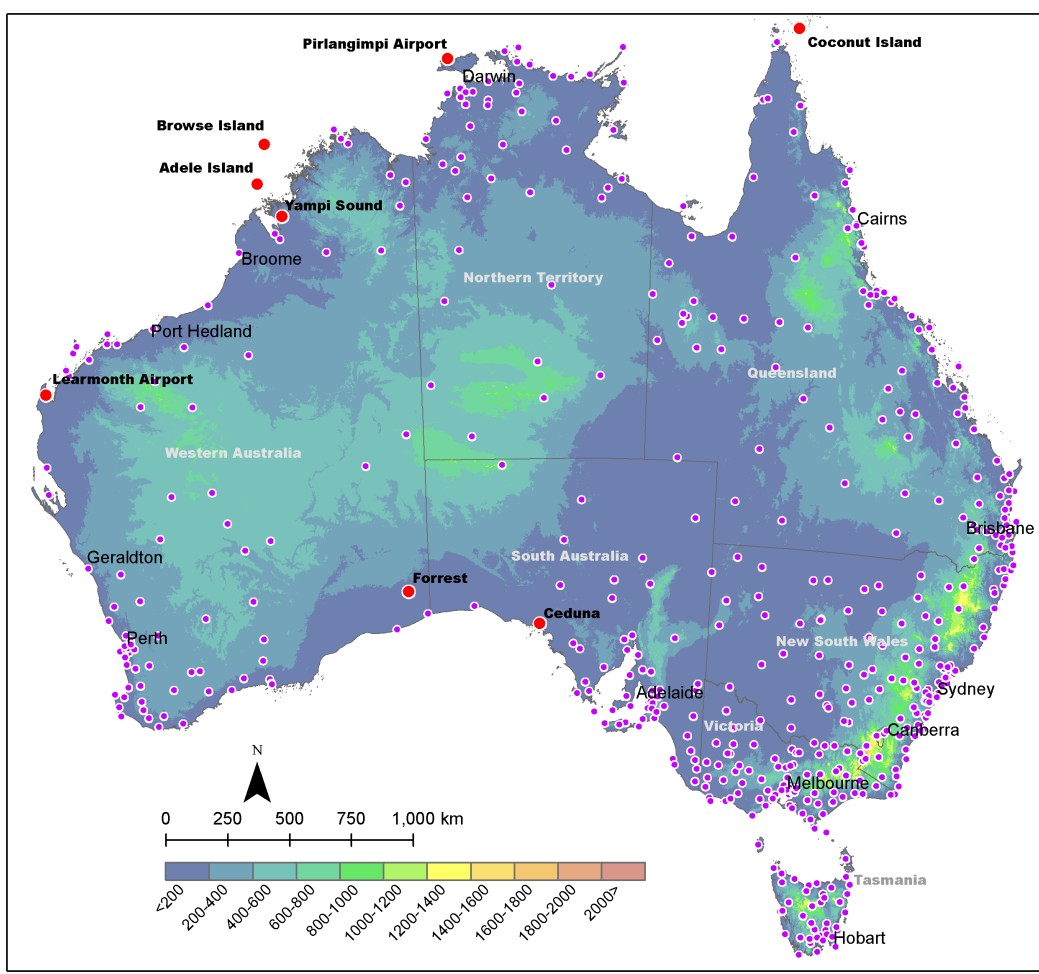

**Figure 2 Elevation map of Australia with locations of major towns/cities and Bureau of Meteorology (BoM) automatic weather stations (AWS).** Purple dots illustrate AWS locations. Red dots denote locations of notable AWS sites (refer to "Results" section). 

In this study, the *Fields* statistical package (*Nychka et al., 2017*) was used to implement the TPS algorithm in R software (*R Development Core Team, 2015*). To guide the mapping of $T_a$, the 9-second Digital Elevation Model (DEM) was used (*Hutchinson et al., 2008*). This was reprojected to Geocentric Datum of Australia 94, Geoscience Australia Lambert projection; and resampled to a spatial resolution of 286 m (roughly equivalent to the spatial resolution of original 9-second DEM). The geographical coordinates of the AWS site locations were then spatially intersected with the newly resampled DEM. This operation provided a consistent template to routinely form TPS models using the AWS observations as data points to the algorithm (on a 30-min basis). Thus, $T_a$ predictions generated by each TPS model were spatially interpolated using the DEM as the *z* variable, along with the coordinate parameters of the inherent cell properties of the DEM acting as the latitude (*x*) and longitude (*y*) variables. This allowed the spline smoothing parameter to be applied continuously across the geographic feature space of the DEM, resulting in a final mapped prediction; saved as GeoTIFF rasters.

## Evaluating TPS interpolation

The performance of the TPS algorithm was evaluated in the period from 1 March 2019 to 29 February 2020. A leave-one-out cross-validation procedure was employed for each AWS site, similar to the method employed in *Webb, Kidd & Minasny (2020)*. Specifically, the training dataset was split into $i$ parts such that $i$ is equal to the number of AWS sites, that is 534. For each AWS in $i$, the $i$th AWS site was kept for validation (i.e. using actual recordings from the evaluation period), while the remaining dataset, comprising of the remaining BoM recordings was used for TPS modelling to predict $T_a$ at the $i$th AWS site. This was performed for each 30-min interval (h) in the evaluation period to produce a set of modelled TPS estimates vs. actual AWS recordings at each site. This equated to 17,567 modelled TPS predictions where observed $T_a$—recorded from each corresponding AWS site—could then be compared. Validation metrics used to assess the modelling accuracy against the $T_a$ recordings, as per *Webb, Kidd & Minasny (2020)*, included the mean absolute error (MAE), root-mean-square error (RMSE), coefficient of determination ($R^2$) and the concordance coefficient. The concordance coefficient ($P_c$) was used to assess agreement between TPS predictions $x$; and actual recordings $y$; that fall on the 45° line through the origin, as defined by *Lin (1989)*:

$$p_c = \frac{2p\sigma_x\sigma_y}{\sigma_x^2 + \sigma_y^2 + (\mu_x - \mu_x)^2}$$

where for $\mu_x$ and $\mu_x$ represent the means for $x$ and $y$, respectively, $\sigma_x^2$ and $\sigma_y^2$ represent the corresponding variances, and $p$ is the correlation coefficient between $x$ and $y$. A concordance rating close to one indicates strong agreement between predicted and actual $T_a$ pairings that fall on the 45° line through the origin.

## Application to real-time monitoring of $T_a$

The proposed methodology, as advocated by *Webb, Kidd & Minasny (2020)*, was adopted in this study for operational real-time monitoring of $T_a$ across continental Australia. However, since the formation of BoM grids and calibration equations were not required in this study, the methodology was retrofitted to consist of two major components. Firstly, the import of 'live' $T_a$ data via the internet from the BoM website, and secondly, the mapping of the observations using TPS interpolation. This was trialled over a 21-day period from 1 June 2020 to 21 June 2020, using real-time BoM observations to drive the system which was fully automated using software R (*R Development Core Team, 2015*). Because new BoM observations are typically available every 30-min, individual AWS site observations were downloaded at this frequency from the BoM observations portal as comma delimited text files (e.g. http://www.bom.gov.au/fwo/IDT60801/IDT60801. <stationIDnumber>.axf). Thus, the mapping system was programmed to query and import recordings every 30-min (bi-hourly) that corresponded to the nearest half-hour at 0- and 30 min (past the hour). Because of telemetry and associated processing delays (observation updates varied from station to station), the system was programmed to make queries at 5, 10, 15, 20, 25 and 30 min within their 30-min processing window. In addition,

a threshold was set where at least 480 out of the 534 BoM stations (i.e. 90% of total available AWS sites that were used in the evaluation analysis) have available observations before the mapping was allowed to commence in their respective processing window. This served to limit the number of missing observations that could otherwise produce significant inaccuracies in the subsequent mapped product. However, if this threshold was not met during the allocated query times, the mapping was still permitted to commence at the 30-min mark regardless of the number of observations available (this was subsequently recorded). To provide context of the proposed system, the same procedure by *Webb, Kidd & Minasny (2020)* found that most observations tended to be imported at the 15-min mark (from the nearest observation hour) with corresponding TPS maps completed thereafter at the 17-min mark. The rationale for this study assumes a similar time frame, albeit at bi-hourly intervals, where observations are imported every 30-min (with an import time lag of ~15-min from the nearest half-hour), followed by $T_a$ mapping thereafter. Note that all AWS recording times in this study were standardised to Australian Eastern Standard Time (AEST).

To interpolate the TPS predictions, the processing schema described in *Webb, Kidd & Minasny (2020)* was used. This consisted the Raster package (*Hijmans & Van Etten, 2012*) in combination with the *Fields* statistical package (*Nychka et al., 2017*) using software R (*R Development Core Team, 2015*), to map and subsequently visualise the predictions in a continuous manner across Australia. To improve processing speed, the clusterR function within the Raster package was parameterised to host the TPS algorithm, thereby enabling mapping to occur using multi-core processors. In this manner, the mapping system was hosted on a high-end cloud computing Linux platform (Ubuntu 18.04 LTS (Bionic)) constituting 16 virtual CPU cores and 64 GB RAM; made available courtesy of the Australian National eResearch Collaboration Tools and Resources project (NeCTAR). Spatial outputs were saved as individual GeoTIFF raster format at a grid cell resolution of 286 m, that is equivalent to the spatial resolution of the resampled DEM.

## RESULTS

### Assessment of the TPS interpolation procedure

Each of the AWS sites underwent the leave-one-out cross-validation analysis to assess TPS prediction accuracy for $T_a$ in the evaluation period: 1 March 2019 to 29 February 2020. This analysis revealed broad similarities across the seasons with MAE values ranging from 1.2 °C (autumn and summer) to 1.3 °C (winter and spring), and similarly RMSE ranging from 1.6 °C to 1.7 °C (Table 1). The $R^2$ and $P_c$ values were above 0.8 indicating that the TPS predictions were strongly correlated to the validation data in addition to being highly associated with the 45° line through the origin (*Lin, 1989*). This assessment also implied that predictions were relatively consistent across the evaluation period and did not vary substantially on a seasonal basis. Moreover, it implied that the TPS interpolation was more suited to predicting $T_a$ in autumn which tended to exhibit superior statistics across all validation measures when compared to the other seasons. This was particularly evident regarding $R^2$ and $P_c$, which registered the highest values of 0.91 and 0.94, respectively. However, TPS predictions tended to be least accurate in spring and winter

**Table 1 Validation statistics for the TPS interpolation procedure showing $R^2$, $P_c$, MAE (°C) and RMSE (°C) values—averaged for each AWS site according to the season.**

|  | Summer | Autumn | Winter | Spring |
|---|---|---|---|---|
| $R^2$ |  |  |  |  |
| mean | 0.89 | 0.91 | 0.86 | 0.91 |
| min | 0.05 | 0.02 | 0.01 | 0.01 |
| max | 0.99 | 0.99 | 0.97 | 0.99 |
| sd | 0.11 | 0.09 | 0.11 | 0.09 |
| $P_c$ |  |  |  |  |
| mean | 0.92 | 0.94 | 0.92 | 0.93 |
| min | 0.18 | 0.14 | 0.18 | 0.09 |
| max | 0.99 | 0.99 | 0.99 | 0.99 |
| sd | 0.09 | 0.08 | 0.09 | 0.08 |
| MAE |  |  |  |  |
| mean | 1.2 | 1.2 | 1.3 | 1.3 |
| min | 0.5 | 0.5 | 0.5 | 0.6 |
| max | 3.2 | 2.8 | 3.3 | 3.6 |
| sd | 0.4 | 0.4 | 0.5 | 0.5 |
| RMSE |  |  |  |  |
| mean | 1.6 | 1.6 | 1.7 | 1.7 |
| min | 0.6 | 0.7 | 0.7 | 0.8 |
| max | 4.3 | 3.5 | 4.3 | 4.3 |
| sd | 0.5 | 0.5 | 0.6 | 0.6 |

**Note:**
sd, standard deviation; min, minimum; max, maximum.

which had MAE and RMSE values greater by 0.1 °C, when compared to the corresponding MAE and RMSE values in autumn. Interestingly, although spring exhibited comparatively inferior MAE and RMSE values, the $R^2$ statistics were similar, both registering 0.91. This suggests that while errors were comparatively larger in spring, they were still very highly correlated to the validation data. However, it should be noted that the coefficient of determination may have been unrealistically overestimated for spring since the seasonal data signal was not removed prior to analysis, as advocated in *Jeffrey et al. (2001)*.

When looking at the histogram distribution of the MAE and RMSE it was apparent that spring and winter had a notable proportion of AWS sites that exhibited values above 2 °C (Fig. 3). This contributed to the inflated error values shown in Table 1. Specifically, spring and winter both had a total of 42 and 46 AWS sites that registered MAE above 2 °C, compared to 22 and 16 AWS sites for summer and autumn, respectively. Similarly, spring and winter also had a large proportion of RMSE values above 2 °C with 147 and 134 AWS sites, respectively, compared to 95 and 91 AWS sites for summer and autumn, respectively. In regard to $R^2$ and $P_c$, winter had a greater proportion of AWS sites that exhibited moderate to weak correlation (≤0.7) with 45 and 32 sites, respectively; compared to 37 and 17 in summer, 11 and 10 in spring, and 13 and 8 in autumn. The high

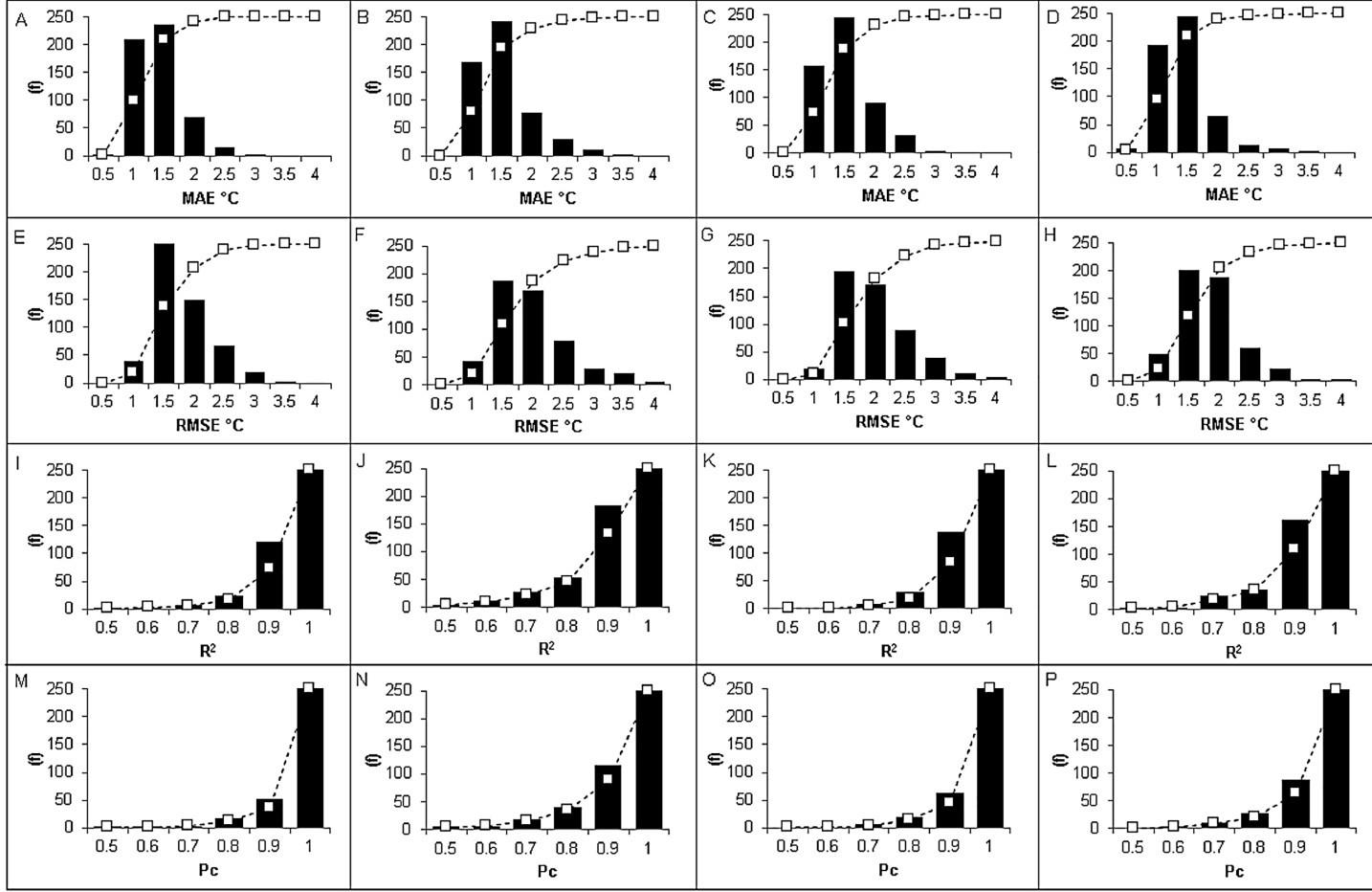

**Figure 3 Histogram plots of validation metrics for MAE, RMSE, $R^2$ and $P_c$ according to each season.** (A) MAE in autumn. (B) MAE in winter. (C) MAE in spring. (D) MAE in summer. (E) RMSE in autumn. (F) RMSE in winter. (G) RMSE in spring. (H) RMSE in summer. (I) $R^2$ in autumn. (J) $R^2$ in winter. (K) $R^2$ in spring. (L) $R^2$ in summer. (M) $P_c$ in autumn. (N) $P_c$ in winter. (O) $P_c$ in spring. (P) $P_c$ in summer.

proportion of low $R^2$ values in winter contributed to the lowest $R^2$ value of 0.86, compared to 0.89, 0.91 and 0.91 for summer, autumn and spring, respectively (Table 1).

When viewing the errors spatially, it was clear that most of the larger interpolation errors transpired in regions where there was a lack of neighbouring AWS sites (Fig. 4). Specifically, the central and western interior parts of Australia tended to exhibit MAE and RMSE values above 2 °C, compared to the eastern half where temperatures were consistently predicted within 2 °C of the actual $T_a$. Of note was the predominately high errors encountered for the coastal areas of Western Australia (between Geraldton and Port Hedland) during summer and spring where the MAE and RMSE prediction errors regularly exceeded 2.5 °C. For example the Learmonth Airport AWS site (Fig. 2) in spring had MAE and RMSE of 3.4 °C and 4.3 °C, respectively, in addition to summer with corresponding MAE and RMSE of 3.2 °C and 4.2 °C, respectively. Outside of this cluster, there were also high MAE and RMSE values for individual AWS sites located at Pirlangimpi Airport (Tiwi Islands, Northern Territory) in spring with 3.6 °C and 4.3 °C,

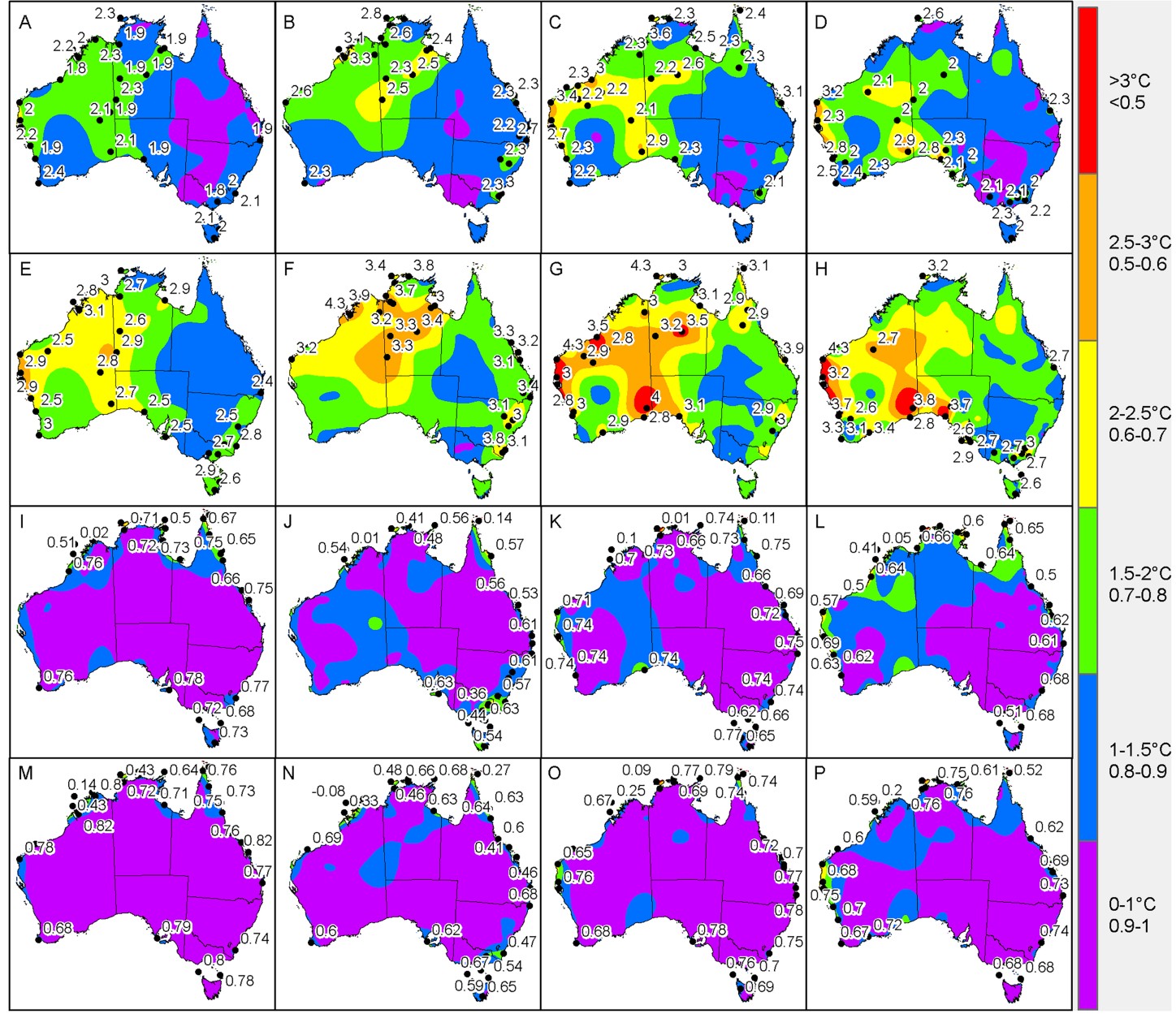

**Figure 4 Interpolated validation metrics (using a two-dimensional smoothing spline) for MAE, RMSE, $R^2$ and $P_c$ according to each season.**
(A) MAE in autumn. (B) MAE in winter. (C) MAE in spring. (D) MAE in summer. (E) RMSE in autumn. (F) RMSE in winter. (G) RMSE in spring.
(H) RMSE in summer. (I) $R^2$ in autumn. (J) $R^2$ in winter. (K) $R^2$ in spring. (L) $R^2$ in summer. (M) $P_c$ in autumn. (N) $P_c$ in winter. (O) $P_c$ in spring.
(P) $P_c$ in summer. Black dots within each panel denote AWS sites where values are above the 95th percentile (labelled with their corresponding value).

respectively; Forrest in Western Australia during summer with corresponding MAE and RMSE of 2.9 °C and 3.8 °C, respectively; and Yampi Sound in the Northern Territory during winter with MAE and RMSE of 3.3 °C and 4.3 °C, respectively. Furthermore, in winter there was a notable cluster of high MAE values emanating from central Australia through to the coastal fringes of Northern Territory and Western Australia (i.e. Darwin through to Broome) with MAE and RMSE consistently above 2 °C.

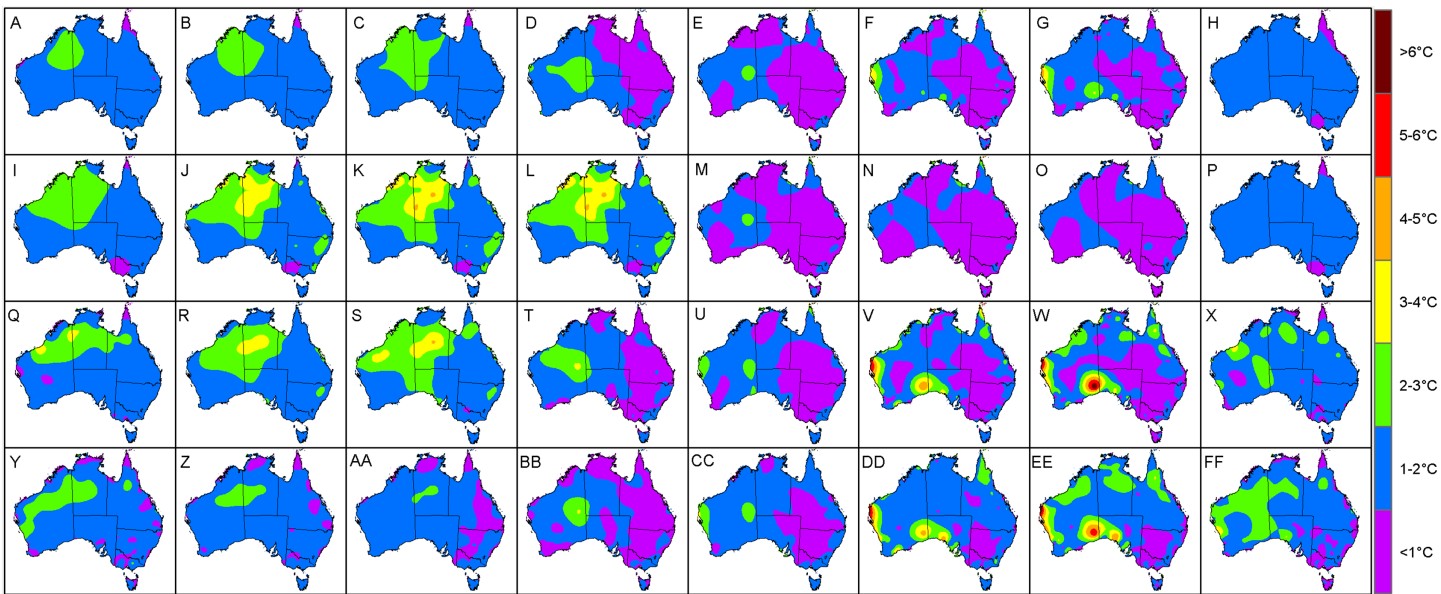

**Figure 5 Interpolated MAE values (using a two-dimensional smoothing spline) according to each season and time of day.** (A) Autumn, 12 am.
(B) Autumn, 3 am. (C) Autumn, 6 am. (D) Autumn, 9 am. (E) Autumn, 12 pm. (F) Autumn, 3 pm. (G) Autumn, 6 pm. (H) Autumn, 9 pm. (I) Winter,
12 am. (J) Winter, 3 am. (K) Winter, 6 am. (L) Winter, 9 am. (M) Winter, 12 pm. (N) Winter, 3 pm. (O) Winter, 6 pm. (P) Winter, 9 pm.
(Q) Spring, 12 am. (R) Spring, 3 am. (S) Spring, 6 am. (T) Spring, 9 am. (U) Spring, 12 pm. (V) Spring, 3 pm. (W) Spring, 6 pm. (X) Spring, 9 pm.
(Y) Summer, 12 am. (Z) Summer, 3 am. (AA) Summer, 6 am. (BB) Summer, 9 am. (CC) Summer, 12 pm. (DD) Summer, 3 pm. (EE) Summer,
6 pm. (FF) Summer, 9 pm.                      

In terms of the $R^2$ and $P_c$, low values tended to emanate along the coastal regions, particularly for Western Australia, Northern Territory and North Queensland coastal regions and neighbouring islands (Fig. 4). For example the lowest values were encountered for sites Pirlangimpi Airport, Browse Island and Coconut Island in spring with $R^2$ of 0.01, 0.1 and 0.1, respectively, and $P_c$ of 0.09, 0.25 and 0.23, respectively. Summer also encountered low $R^2$ of 0.05, 0.05 and 0.31, respectively, with corresponding $P_c$ of 0.18, 0.2 and 0.52. The same sites in winter also had the lowest $R^2$ of 0.41, 0.01 and 0.14, respectively, along with corresponding $P_c$ of 0.48, 0.08 and 0.27.

When observing MAE (Fig. 5) and RMSE (Fig. 6) over a 24-h period, it was clear that the high values encountered for the coastal areas of Western Australia in summer and spring tended to occur during afternoons. Specifically, these had MAE and RMSE ranging between 4 °C and 6 °C for times 3 pm to 6 pm, that is 1 pm to 4 pm, Australian Western Standard Time (AWST). Of note was the Learmonth Airport AWS site registering MAE of 6.9 °C and RMSE of 7.6 °C, peaking at 5 pm (3 pm, AWST) in summer (Fig. 7). Similarly, very high error values were encountered for the south-eastern area of Western Australia in spring, notably for the Forrest AWS site at 6 pm, which registered 6.1 °C and 6.3 °C for MAE and RMSE, respectively. This was in addition to 5.3 °C and 7 °C, respectively, for the same site in summer, along with the Ceduna AWS site (South Australia) at 6 pm, registering high MAE and RMSE values of 5.4 °C and 6 °C, respectively. During winter the trend for high MAE and RMSE emanating from central

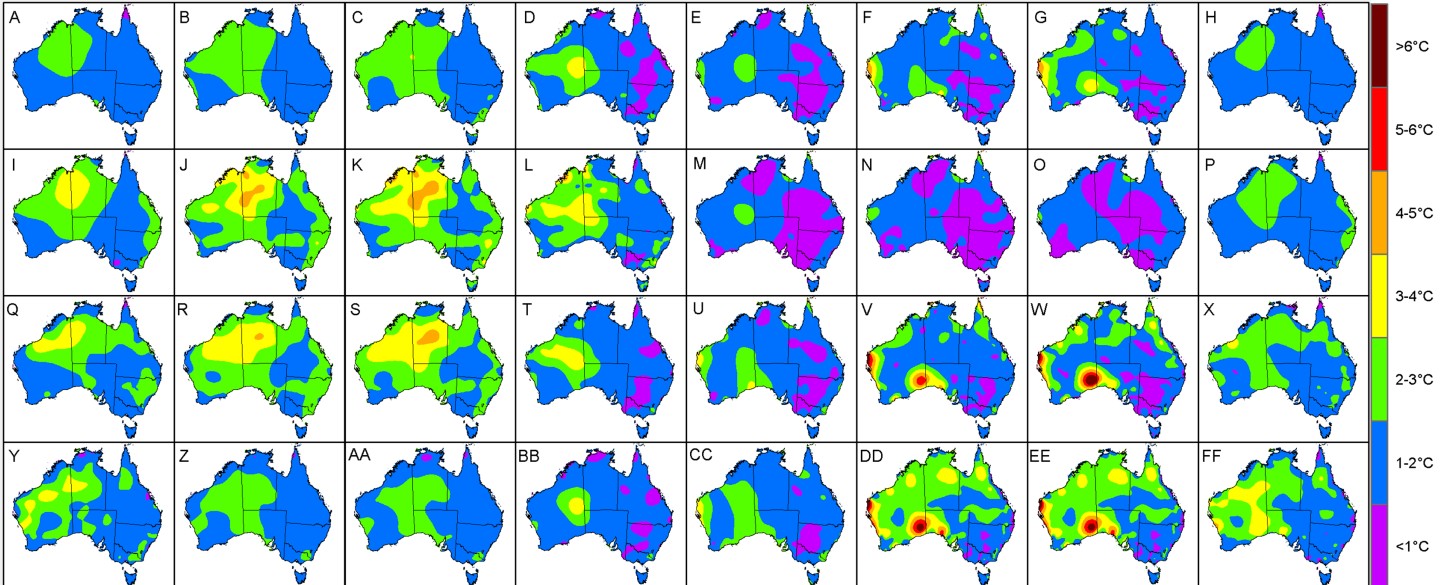

**Figure 6 Interpolated RMSE values (using a two-dimensional smoothing spline) according to each season and time of day.** (A) Autumn, 12 am. (B) Autumn, 3 am. (C) Autumn, 6 am. (D) Autumn, 9 am. (E) Autumn, 12 pm. (F) Autumn, 3 pm. (G) Autumn, 6 pm. (H) Autumn, 9 pm. (I) Winter, 12 am. (J) Winter, 3 am. (K) Winter, 6 am. (L) Winter, 9 am. (M) Winter, 12 pm. (N) Winter, 3 pm. (O) Winter, 6 pm. (P) Winter, 9 pm. (Q) Spring, 12 am. (R) Spring, 3 am. (S) Spring, 6 am. (T) Spring, 9 am. (U) Spring, 12 pm. (V) Spring, 3 pm. (W) Spring, 6 pm. (X) Spring, 9 pm. (Y) Summer, 12 am. (Z) Summer, 3 am. (AA) Summer, 6 am. (BB) Summer, 9 am. (CC) Summer, 12 pm. (DD) Summer, 3 pm. (EE) Summer, 6 pm. (FF) Summer, 9 pm.

Australia and the coastal fringes of Northern Territory and Western Australia tended to occur during early mornings from 3 am to 9 am (1 am to 7 am, AWST), with MAE and RMSE ranging 3–5 °C. The AWS sites with the greatest error in these parts were Adele Island and Yampi Sound which both registered a MAE of 6.4 °C and RMSE of 6.4 °C and 6.7 °C, respectively (Figs. 5 and 6). Both sites are located in the northern coastal region of Western Australia (NB: the locations of all aforementioned AWS sites are depicted in Fig. 2).

Regarding $R^2$ and $P_c$, it was revealed that low values <0.5 were mostly evident during late nights and early mornings, particularly during winter and summer (Figs. 8 and 9). This tended to be prevalent throughout the central western interior and the coastal fringes of Western Australia, Northern Territory and North Queensland. Specifically, $R^2$ and $P_c$ <0.5 tended to occur from midnight through to 6 am, suggesting that the TPS predictions were not highly correlated with the validation data during these times. This occurred despite the RMSE and MAE registering values below 2 °C for some sites. For example the AWS site at Browse Island during summer at 12 am registered $R^2$ and $P_c$ values of 0.04 and 0.2, respectively, while MAE and RMSE was 0.9 °C and 1.1 °C, respectively (Fig. 7). This was also encountered for the Coconut Island AWS during winter mornings, for example at 9 am where the $R^2$ and $P_c$ values registered 0.0 and 0.1, respectively, while the MAE and RMSE was 1.4 °C and 1.7 °C, respectively. This suggested that while predictions were reasonably accurate, there was little to no correlation with the validation data.

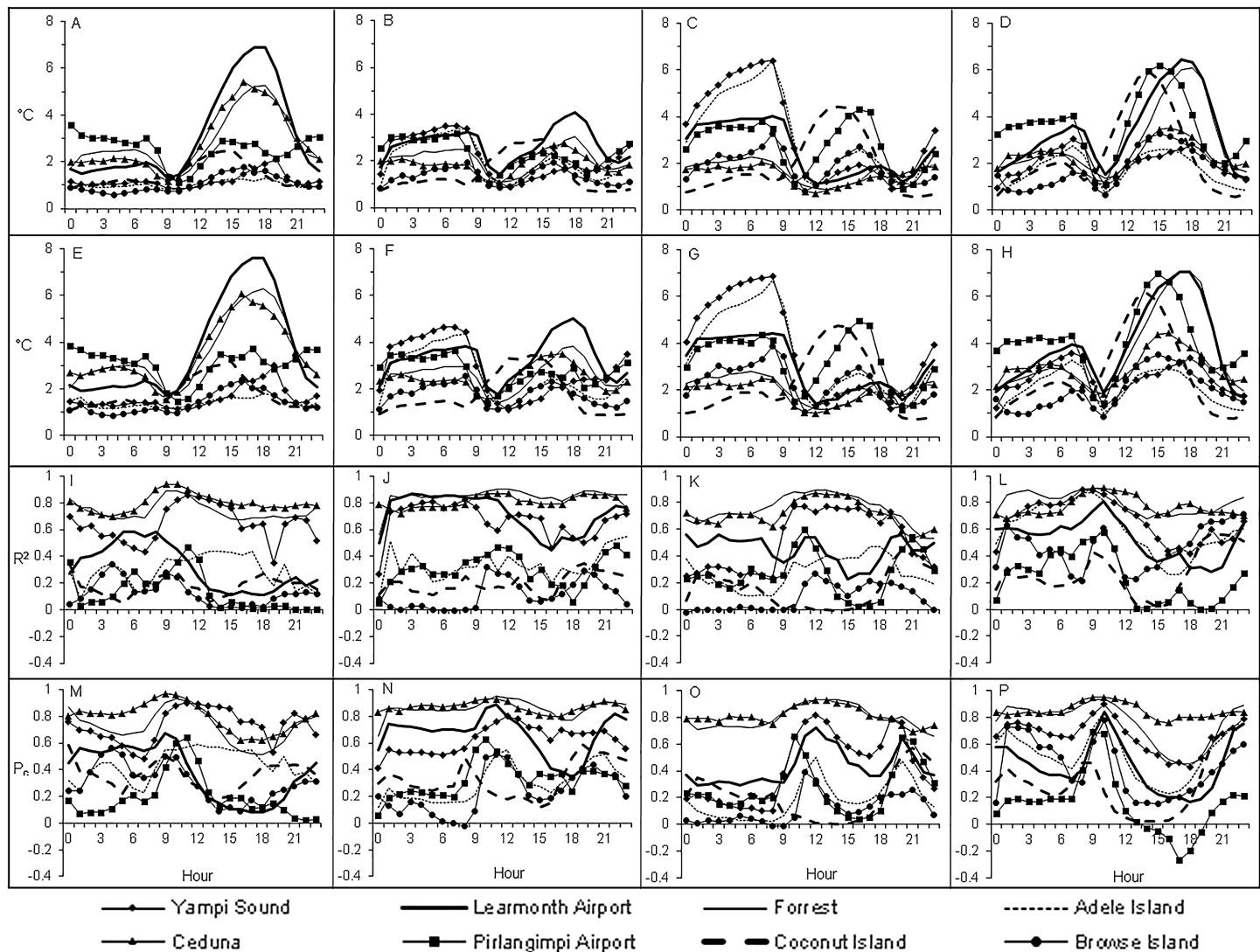

**Figure 7 Line plots for selected AWS sites over a 24-h period in each season for validation metrics concerning MAE, RMSE, $R^2$ and $P_c$.**
(A) MAE in autumn. (B) MAE in winter. (C) MAE in spring. (D) MAE in summer. (E) RMSE in autumn. (F) RMSE in winter. (G) RMSE in spring. (H) RMSE in summer. (I) $R^2$ in autumn. (J) $R^2$ in winter. (K) $R^2$ in spring. (L) $R^2$ in summer. (M) $P_c$ in autumn. (N) $P_c$ in winter. (O) $P_c$ in spring. (P) $P_c$ in summer.

## Assessment of mapping $T_a$ in near real-time

The TPS methodology was applied to mapping $T_a$ in real-time at 30-min intervals over a 21-day period from 1 June 2020 to 21 June 2020. This exercise resulted in 1007 maps being produced which aligned to the total number of 30-min processing intervals in the trial period; confirming all possible maps were successfully processed. On analysing the map completion times, the majority of the maps were completed at 28-min (Fig. 10). Specifically, 410 and 414 maps were produced for their respective 0- and 30-min processing intervals. This corresponded directly to the AWS import times (Fig. 11), with the same proportion of AWS observations reaching the 480-observation threshold import limit at the 15-min mark; thereby permitting $T_a$ mapping to commence. Thus, import

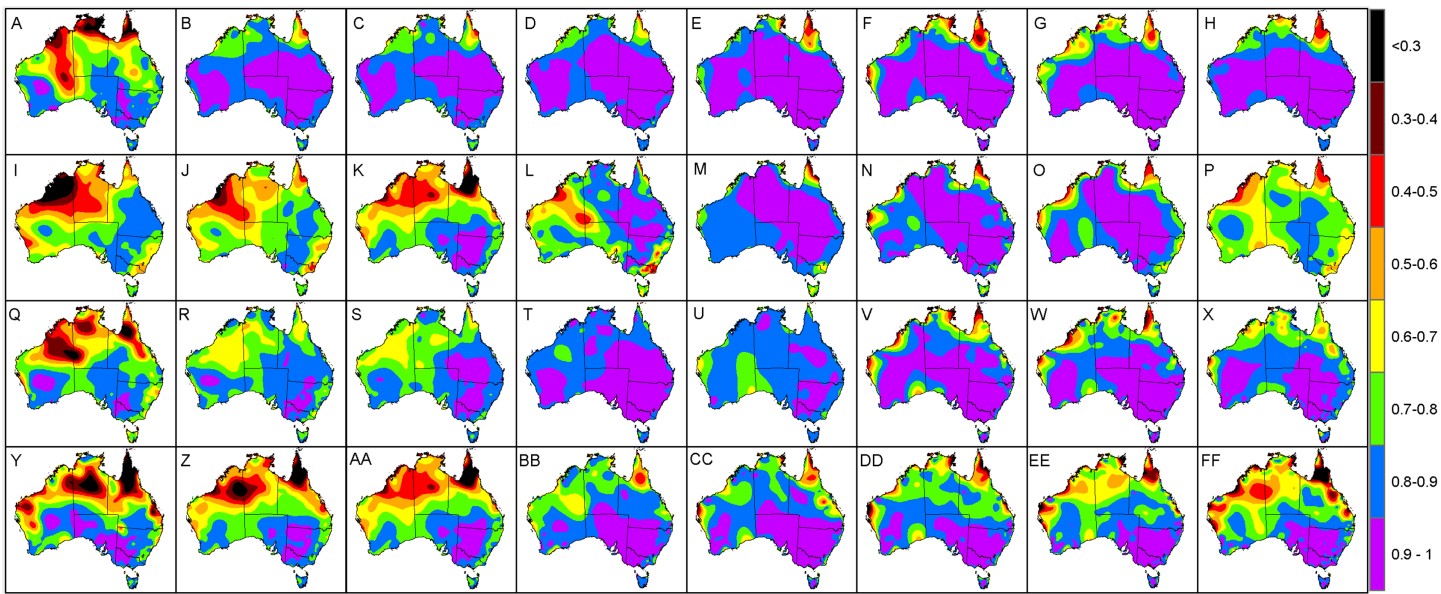

**Figure 8 Interpolated $R^2$ values (using a two-dimensional smoothing spline) according to each season and time of day.** (A) Autumn, 12 am. (B) Autumn, 3 am. (C) Autumn, 6 am. (D) Autumn, 9 am. (E) Autumn, 12 pm. (F) Autumn, 3 pm. (G) Autumn, 6 pm. (H) Autumn, 9 pm. (I) Winter, 12 am. (J) Winter, 3 am. (K) Winter, 6 am. (L) Winter, 9 am. (M) Winter, 12 pm. (N) Winter, 3 pm. (O) Winter, 6 pm. (P) Winter, 9 pm. (Q) Spring, 12 am. (R) Spring, 3 am. (S) Spring, 6 am. (T) Spring, 9 am. (U) Spring, 12 pm. (V) Spring, 3 pm. (W) Spring, 6 pm. (X) Spring, 9 pm. (Y) Summer, 12 am. (Z) Summer, 3 am. (AA) Summer, 6 am. (BB) Summer, 9 am. (CC) Summer, 12 pm. (DD) Summer, 3 pm. (EE) Summer, 6 pm. (FF) Summer, 9 pm.

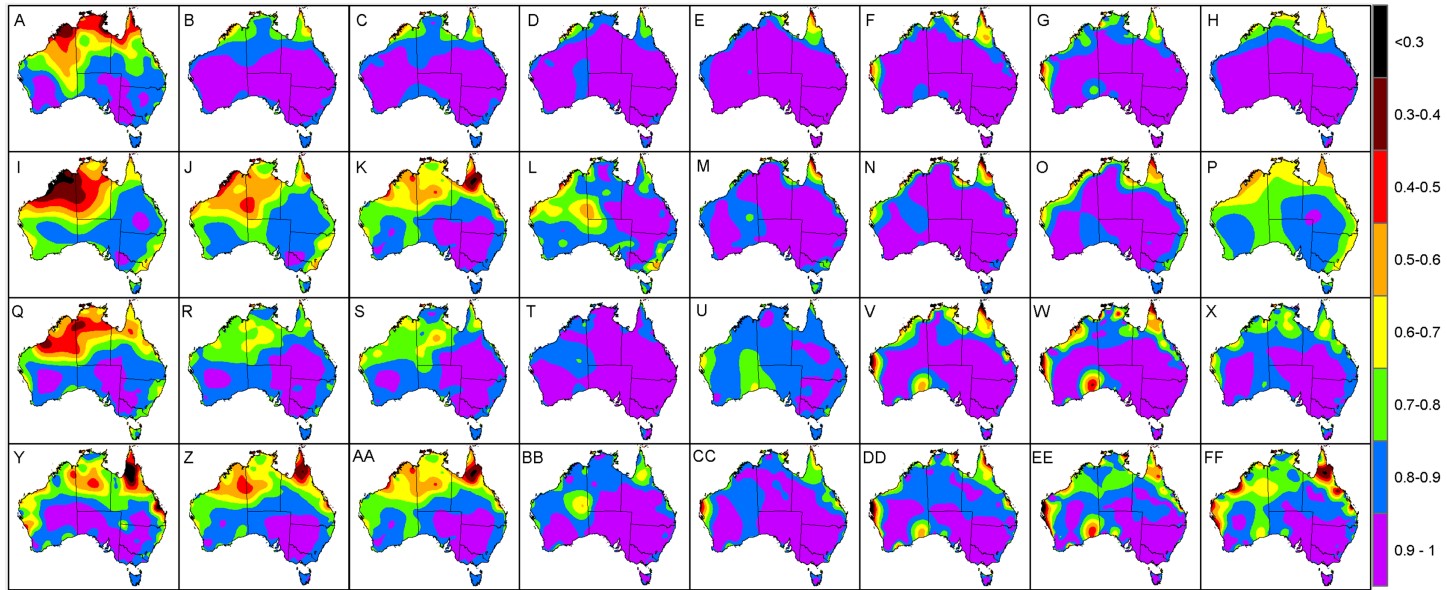

**Figure 9 Interpolated $P_c$ values (using a two-dimensional smoothing spline) according to each season and time of day.** (A) Autumn, 12 am. (B) Autumn, 3 am. (C) Autumn, 6 am. (D) Autumn, 9 am. (E) Autumn, 12 pm. (F) Autumn, 3 pm. (G) Autumn, 6 pm. (H) Autumn, 9 pm. (I) Winter, 12 am. (J) Winter, 3 am. (K) Winter, 6 am. (L) Winter, 9 am. (M) Winter, 12 pm. (N) Winter, 3 pm. (O) Winter, 6 pm. (P) Winter, 9 pm. (Q) Spring, 12 am. (R) Spring, 3 am. (S) Spring, 6 am. (T) Spring, 9 am. (U) Spring, 12 pm. (V) Spring, 3 pm. (W) Spring, 6 pm. (X) Spring, 9 pm. (Y) Summer, 12 am. (Z) Summer, 3 am. (AA) Summer, 6 am. (BB) Summer, 9 am. (CC) Summer, 12 pm. (DD) Summer, 3 pm. (EE) Summer, 6 pm. (FF) Summer, 9 pm.

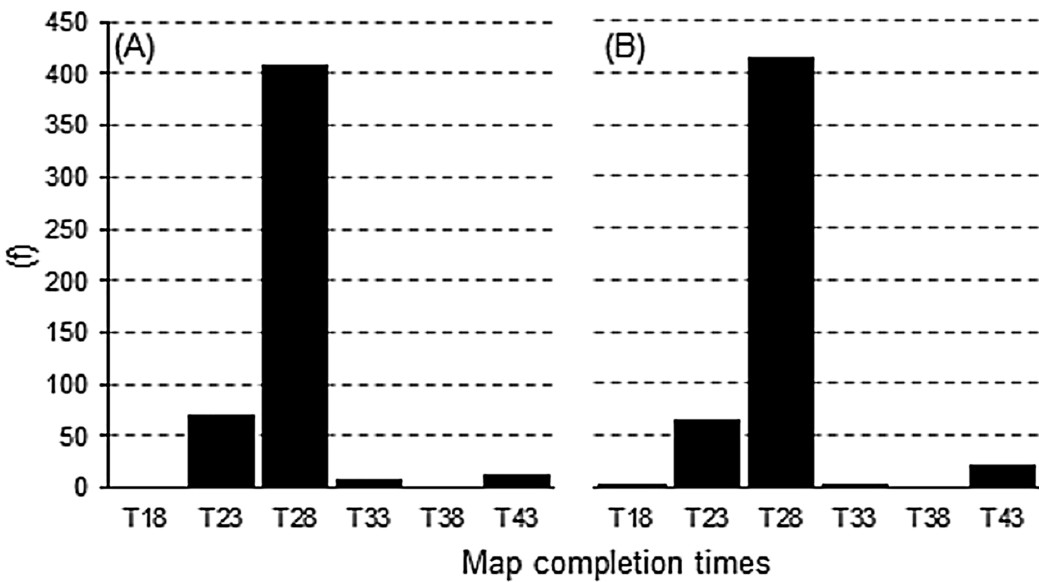

**Figure 10 Frequency of map completion times (minutes from AWS observation time, T) in accordance to their bi-hourly processing intervals.** (A) 0 min. (B) 30 min.

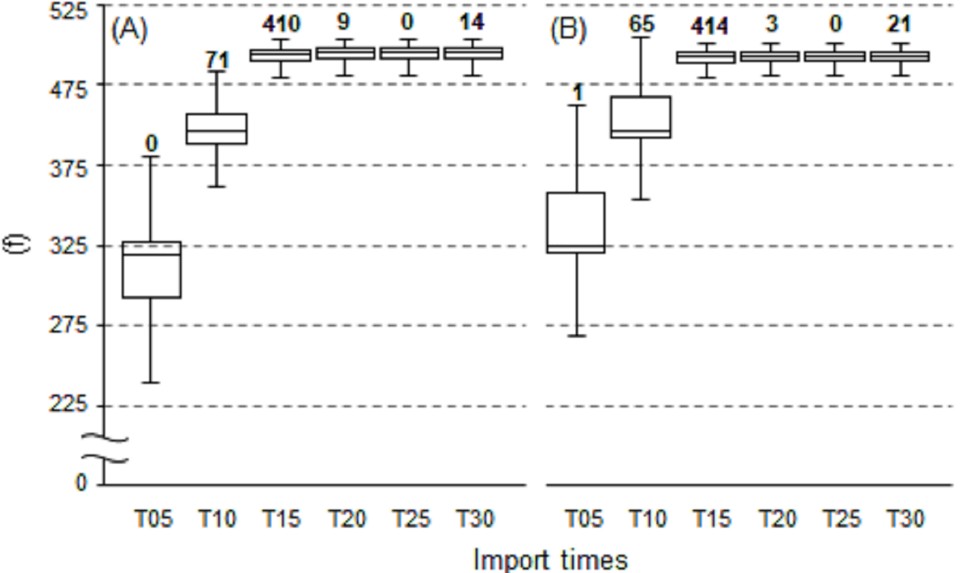

**Figure 11 Box and whisker plots for AWS import times (minutes from AWS observation time, T) in accordance to their bi-hourly processing intervals.** (A) 0 min. (B) 30 min. Numbers in bold denote frequencies when the 480-observation threshold limit was reached.

times that occurred at 15-min, equated to resulting maps being completed at 28-min from the AWS observation time. From this, it can be deduced that on all occasions the map processing time was 13-min, regardless of the interval being processed. It should be noted that on 35 occasions the 480-observation threshold limit was not reached, resulting in

maps—that did not meet this criterion—being produced at the 30-min mark. This equated to 14 and 21 maps produced at the 0- and 30-min processing intervals, respectively.

## DISCUSSION

### Appraisal of the TPS interpolation procedure

On the whole, the TPS interpolation method was a reliable predictor of $T_a$ across Australia with an RMSE of 1.65 °C, that is when averaged across the seasons (Table 1). When compared to previous studies, this error was similar to *Jeffrey et al. (2001)* with RMSE of 1.5 °C and 1.9 °C for daily maximum and minimum temperatures, respectively; and *Jones, Wang & Fawcett (2009)* with corresponding RMSE of 1.2 °C and 1.7 °C. On a seasonal basis the TPS predictions tended to be least accurate in spring and winter which had MAE and RMSE values larger by 0.1 °C compared to the same measures in autumn. When viewing these errors spatially, it was clear that the majority of the larger interpolation errors transpired in the central and western interior parts of Australia. This is unsurprising given the station density in these parts are relatively sparse in addition to large temperature variances which tend to produce inflated errors (*Jeffrey et al., 2001*; *Jones & Trewin, 2000*). Of note was the predominately high errors encountered for the coastal areas of Western Australia (between Geraldton and Port Hedland) during summer and spring afternoons where prediction errors were regularly above 2.5 °C. This was in addition to high MAE values for individual AWS sites located at Forrest in Western Australia and Ceduna in South Australia. Collectively, these regions tend to experience very strong temperature gradients, particularly concerning maximum temperatures, since their proximity between the coast and inland deserts result in local climate regimes being invariably affected by the relatively cool ocean to the west and hot desert interior to the east (*Jones, Wang & Fawcett, 2009*). These are increasingly difficult to model with a sparse network of observation sites since these errors tended to be amplified during mid to late afternoons in late spring and summer when the temperature gradients were at their peak. Also, temperatures in these areas vary considerably over short periods leading to a tendency for larger errors (*Jones & Trewin, 2000*).

Concerning winter, the trend for high MAE and RMSE in central Australia and coastal fringes of Northern Territory and Western Australia tended to occur during early mornings from 3 am to 9 am (1 am to 7 am, AWST). As acknowledged previously, the accuracy of the mapping was limited in these regions due to an insufficient network of AWS sites. Also, AWS sites in the coastal fringes tend to have tight climate gradients as a result of local maritime effects (*Jones, Wang & Fawcett, 2009*). This possibly contributed to the low $R^2$ and $P_c$ values encountered in Figs. 4, 8 and 9—despite some AWS sites exhibiting relatively small MAE and RMSE values, for example Browse Island and Coconut Island AWS sites (Fig. 7). Thus, the predictions tended to be highly variable over a narrower prediction range due to the tighter temperature gradient in these climates. Combined with a sparse network of AWS sites, the TPS method was unable to account for this on a sub-hourly timescale. Moreover, the spread of AWS sites in remote coastal locations—for example Adele Island, Yampi Sound and Pirlangimpi Airport AWS sites— tend to have considerably larger errors as a result of unique and often complex

microclimates, thereby compounding the variability (*Jones, Wang & Fawcett, 2009*). It should also be noted that the larger errors for the central interior parts of Australia may also be due to the weaker link between altitude and temperature – for which the TPS algorithm is reliant (*Hutchinson, 1991*). This is because minimum temperatures have a highly variable and complex relationship with topography for which elevation and its association with lapse rates are only one part (*Rolland, 2003*; *Trewin, 2005*). Considering minimum temperatures tend to transpire during early mornings—as encountered for AWS sites in winter (Figs. 5 and 6)—a multivariate approach to modelling might be more appropriate along with a denser network of AWS sites. This approach was conducted by *Webb, Kidd & Minasny (2020)*, and showed errors improved during winter when using regression tree interpolation (along with multiple terrain and satellite covariate datasets). However, the substantially longer processing times may not be appropriate for real-time application, negating its ability to produce outputs in a timely manner as required for this study. Similar experiments contrasting TPS, ordinary kriging and inverse distance weighting interpolation found that TPS was more accurate and required few guiding covariates (*Jarvis & Stuart, 2001b*). This justified the selection of the TPS method in the current study, even though kriging can be an equally effective method (*Hutchinson, 1991*). However, kriging requires considerable computational overhead (*Jarvis & Stuart, 2001b*) and therefore, in the context of this study, not ideal for real-time application.

It should be commented that the cross-validation analysis adopted in this study would likely overestimate the error since predictions were made at locations that have actual data observations. This would be less of a concern for regions where the number of observation points is numerous, such as for the majority of land areas in south-east Australia—which tended to have more accurate $T_a$ predictions compared to the western interior. Nevertheless, this exemplifies that the sparse network of AWS sites in central and western coastal areas of Australia was a notable factor contributing to larger interpolation errors. It should be further commented that while the cross-validation analysis was valid using a static dataset, in reality and as exemplified during real-time application, interpolation could only commence when the predefined threshold of 480 AWS observations was met. Thus, on most occasions' predictions were based on the minimum allowable number of AWS sites and therefore prone to produce less accurate predictions compared to using an entire dataset. To evaluate this scenario, a K-fold cross validation was implemented (*Hastie, Tibshirani & Friedman, 2009*). Specifically, the training dataset—represented by AWS observations in each 30-min interval (h) within the evaluation period—was split into $K$ equal parts using random sampling. Where the $K$th part was kept for validation and the remaining K-1 part were combined for TPS modelling in each fold. In this manner, the predictions produced by the modelling were assessed against the held back validation subset and was repeated $K$ times, such that each $K$ validation subset was used once to assess the K-1 model. In this study, $K = 10$ was specified, representing 90% of the dataset for TPS modelling and 10% for validation in each fold, that is equivalent to 480 and 54 AWS sites, respectively.

The result of the K-fold analysis (Table 2) revealed broad similarities with Table 1. For example, the MAE of 1.2 °C, 1.2 °C, 1.3 °C and 1.3 °C, respectively for summer,

**Table 2 K-fold cross-validation statistics for the TPS interpolation procedure showing $R^2$, $P_c$, MAE (°C) and RMSE (°C) values—averaged for each season.**

|          | Summer | Autumn | Winter | Spring |
|----------|--------|--------|--------|--------|
| $R^2$    |        |        |        |        |
| mean     | 0.93   | 0.92   | 0.9    | 0.92   |
| min      | 0.4    | 0.43   | 0.04   | 0.41   |
| max      | 0.99   | 0.99   | 0.99   | 0.99   |
| sd       | 0.04   | 0.05   | 0.07   | 0.06   |
| $P_c$    |        |        |        |        |
| mean     | 0.96   | 0.96   | 0.95   | 0.96   |
| min      | 0.62   | 0.62   | 0.19   | 0.63   |
| max      | 1      | 1      | 1      | 1      |
| sd       | 0.02   | 0.03   | 0.04   | 0.03   |
| MAE      |        |        |        |        |
| mean     | 1.2    | 1.2    | 1.3    | 1.3    |
| min      | 0.5    | 0.5    | 0.4    | 0.5    |
| max      | 2.7    | 2.5    | 3      | 3.1    |
| sd       | 0.3    | 0.3    | 0.4    | 0.4    |
| RMSE     |        |        |        |        |
| mean     | 1.7    | 1.6    | 1.7    | 1.8    |
| min      | 0.7    | 0.6    | 0.5    | 0.7    |
| max      | 6.4    | 7.6    | 10     | 7.4    |
| sd       | 0.4    | 0.4    | 0.6    | 0.5    |

**Note:**
   sd, standard deviation; min, minimum; max, maximum.

autumn, winter and spring, and corresponding RMSE of 1.7 °C, 1.6 °C, 1.7 °C and 1.8 °C, respectively, deviated by no more than 0.1 °C when compared to the equivalent measures in Table 1. This indicated that overall the prediction accuracy did not deteriorate greatly when interpolation was based on the minimum allowable number of AWS sites. It also justified that the 480 AWS threshold adopted in this study was an acceptable limit for the real-time application.

## Appraisal of mapping $T_a$ in near real-time and application to digital mapping

The TPS interpolation applied in real-time was capable of producing sub-hourly $T_a$ maps typically within 28-min of the observation being recorded by the available AWS sites (Fig. 10). Specifically, import times were generally reached for the predefined threshold of 480 observations at the 15-min mark (Fig. 11) which was followed by a 13-min processing lag. In this regard, maps were consistently available within their 30-min processing window and had a high degree of temporal reliability—with all possible maps produced in the 21-day trial period. The resulting maps were presented on a digital web mapping platform to allow real-time access and interrogation ability of each output. An example of this application can be accessed at URL http://austemperature.live/ (Fig. 12).

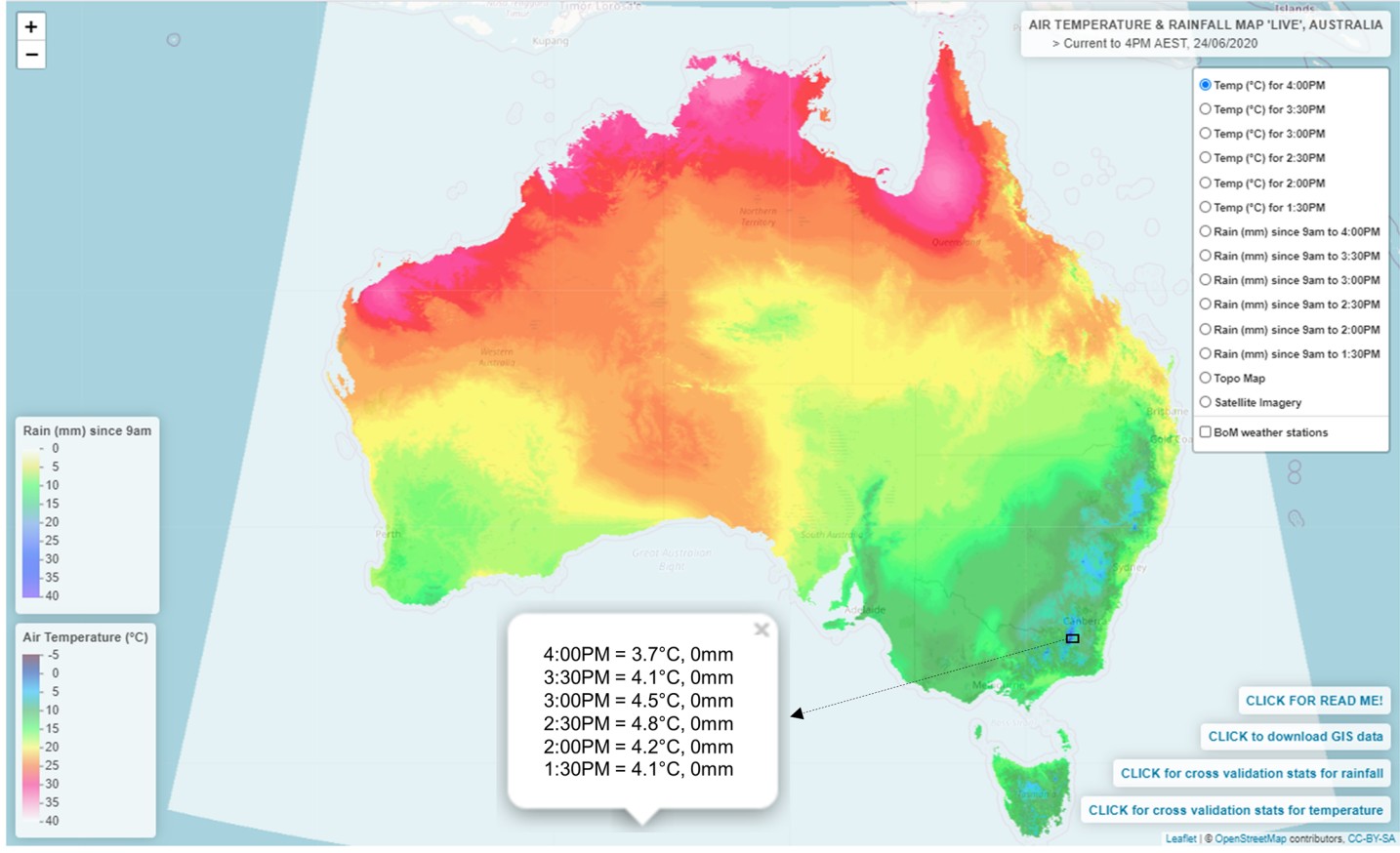

**Figure 12 Web map environment for displaying and spatially interrogating the near real-time $T_a$ outputs. An example can be viewed at URL http://austemperature.live/.** © OpenStreetMap contributors, CC BY SA.               

A GeoServer backend was used to host current outputs to allow geospatial representation and sharing of outputs via a Wep Map Service (*Open Source Geospatial Foundation, 2019*). The maps can be spatially queried to reveal temperatures for the current hour and for the previous 3-h (at 30-min intervals). This is enabled via web application packages *shiny* and *leaflet* (*Chang et al., 2019*; *Cheng, Karambelkar & Xie, 2019*) within the R programming environment (*R Development Core Team, 2015*). In this fashion, maps can be spatially interrogated via an on-the-fly 'data drilling' for any geographical location in Australia (via mouse click). A facility to view the cross-validation statistics of each map output is also provided as well as the ability to download each newly created map for use in GIS applications. A potential new feature is to provide an error map for each subsequent map produced (similar to Figs. 5 and 6). This would provide an approximate error measure for regions with limited AWS sites which tended to be high, as encountered in this study. Note that rainfall mapping outputs are also presented in the application, although this should be used with caution due to the preliminary nature of this work.

## CONCLUSIONS

The methods described in this study were successful for operational real-time spatial mapping of $T_a$ at high spatiotemporal across Australia. The TPS interpolation method was

best suited for mapping $T_a$ during autumn and was comparatively less accurate during winter and spring. In particular, areas, where there was a lack of AWS sites, tended to underperform. These areas included the central and western interior regions of Australia, as well for the north-west coastal areas of Western Australia and parts of the Northern Territory coastline. On a temporal basis, the errors were amplified during the afternoons, particularly around the coastal regions of Western Australia, during spring and summer. In winter, errors tended to be higher in central Australia and the coastal fringes of Northern Territory and Western Australia, from 3 am to 9 am. In terms of applying the TPS method to real-time operational mapping, the mapping system was able to regularly provide spatial outputs within 28-min of AWS site observations being recorded. In addition, it also had a high degree of temporal reliability with all maps produced in the 21-day trial period. Outputs were sequentially displayed on purpose-built web mapping application to exemplify real-time application of the outputs. In this regard, the methodology employed in this study would be highly suited for similar applications requiring real-time processing and delivery of climate data at high spatiotemporal resolutions across a large landmass, suitably complimented with a relatively dense network of observation sites.

### Funding
This research was supported by Tasmanian Partnership for Advanced Computing and by use of the Nectar Research Cloud. The Nectar Research Cloud is a collaborative Australian research platform supported by the National Collaborative Research Infrastructure Strategy (NCRIS). The funders had no role in study design, data collection and analysis, decision to publish, or preparation of the manuscript.

### Grant Disclosures
The following grant information was disclosed by the authors:
Nectar Research Cloud.
National Collaborative Research Infrastructure Strategy (NCRIS).

### Competing Interests
The authors declare that they have no competing interests.

### Author Contributions
- Mathew Webb conceived and designed the experiments, performed the experiments, analysed the data, prepared figures and/or tables, authored or reviewed drafts of the paper, and approved the final draft.
- Budiman Minasny conceived and designed the experiments, analysed the data, authored or reviewed drafts of the paper, provided guidance and supervision for producing the manuscript, and approved the final draft.

## Data Availability

Webb, Mathew, & Minasny, Budiman. (2020). A digital mapping application for quantifying and displaying air temperatures at high spatiotemporal resolutions in near real-time across Australia (Version 1) [Data set]. PeerJ. Zenodo. DOI 10.5281/zenodo.4037541.

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
