# Peer review of "A digital mapping application for quantifying and displaying air temperatures at high spatiotemporal resolutions in near real-time across Australia"

_PeerJ, doi:10.7717/peerj.10106_

## Round 0.1 · original submission · Minor Revisions

Based on the comments from two anonymous reviewers, your article requires minor revisions before being accpepted. Please submit your revision within two or three weeks. Thank you for your selection of PeerJ.

Reviewer 1 ·

Basic reporting

No comment

Experimental design

No comment

Validity of the findings

The criteria for not rejecting an AWS (lines 122-125) suggest that each iteration of the evaluation process the number fo stations actually participating in the TPS procedure would differ. Tamperatures at stations in regions with sparse coverage, could end up being estimated from data coming from quite distant areas, and thus presenting large estimation errors. Could the authors discuss the potential significance of this effect?
Narrowing down the validation period and AWSs used, to those "overall commonly available" might not prove practical. However, an estimate might be obtained by employing an additional completeness criterion based on the percentage of "active" nearby stations during each estimation step.
This could also be relevant to the discussion in lines 203-204 and perhaps lead to an additional comment after line 345 on the possible effect of a reduced number of stations available during the TPS interpolation procedure .

Additional comments

A very interesting article with great potential for application. Proper data handling and analysis, very well writen and presented. Apart from the comment in the "Validity of the findings" section, just a few more, minor, comments/suggestions:
1. Air temperature is measured with an accuracy of one decimal so using two decimals to report the respective MAE might be considered an exaggeration. In that respect, an estimation of the combined uncertainty -instead of simply the MAE- of the estimations made by the method, might also be of interest.
2. There are but a few grammatical errors and missing/misused words (i.e. lines 38, 54, 123, 124, 171, 299)
3. There is a narration gap in lines 42-43, from generaly speaking about the dependence of (environmental) variables on geographical factors to the specifics of air temperature. Perhaps a bridging sentence could be added.
5. Please indicate how many AWS stations were discarded (line 125)
6. Concordance coefficient is indexed as Pc, pc (capital and small p, in the text) and CC in the raw data. Please choose one.
7. The setup of the mapping systems seemed clearly described (lines 196-208), however it was only after lines 288-291 that ts operation was really understood. This is a very important feature of the system. Perhaps the description could be improved and the rationalle behind the overall design, better explained.
8. Line 210: perhaps the authors meant "visualize"
9. Line 243: less than 10% is hardly a large proportion
10. Line 328: "minimum" should be removed from here
11. Raw data provided were easy to identify and use, except for the Excel version of figure 8 (my, older, version of Excel could not display that) and the MAE tiffs (evaluation\data\spatial\mae) that seem to be blank (instead of showing the plates of Figure 5).

Reviewer 2 ·

Basic reporting

no comment

Experimental design

no comment

Validity of the findings

no comment

Additional comments

Comments on “A digital mapping application for quantifying and displaying air temperatures at high spatiotemporal resolutions in near real-time across Australia”

The authors applied the thin plate spline method to interpolation T2m that were observed at a sparse in situ meteorological monitoring network across Australia toward the creation of high resolution T2m maps. The paper is generally well written and the analysis sounds meaningful. However, the technique is a bit outdated given the available of more contemporary yet advanced data fusion and assimilation methods.
1.Line 43-44: The variations in temperature in space is also highly regulated by factors such as aerosols via direct radiative effect (Mitchell et al., 1995, doi:10.1175/1520-0442(1995)008<2364:OSTGGA>2.0.CO;2; Li et al. 2017, doi:10.1093/nsr/nwx117 ), clouds (Xue et al., 2019, doi:10.1007/s00382-018-4505-8), synoptic-scale circulation (Redmond and Koch 1991. doi: 10.1029/91WR00690; Domonkos et al. 2003, doi:10.1002/joc.929; Liu et al., 2018, doi: 10.1175/JCLI-D-17-0608.1), which is suggested to be mentioned at least.
2.Line 67: thin plate smoothing spline performed poorly with limited observations, why didn’t have a try on the widely applied Kriging method?
3.Line 128: as per your description, air temperature should be the commonly applied 2m temperature which should be denoted as T2m to be in line with the common usage in meteorology. Please change the usage throughout the paper.
4.Methodology: currently, especially at the time where both satellite observations and meteorological reanalysis are widely applied, creating a gridded field of geophysical parameters simply using a 2D interpolation method seems to be out of date and less accurate, though it could be a timely way. A popular method is to fuse/assimilate the in situ observations with a given background field either from satellite observations or model output since they may provide more accurate estimates over remote regions with limited or even no in situ observations.
5.Line 248-249: As stated by the authors, larger interpolation errors would be introduced in regions with limited neighboring AWS sites. How to deal with this drawback, or just providing estimations with large uncertainty?
6.Line 263–264: why large errors were observed in the coastal regions during afternoons in summer and spring? Is this due to instrumental bias or region-specific anomaly? Please try to give some explanation.

---

## Round 0.2 · accepted · Accept

Thank you for your submission to the journal of PeerJ. The minor comments proposed by the anonymous reviewers have been well addressed in the revised manuscript, and thus it can be accepted. Look forwad to receiving your next work.